# Predicting Accurate Lagrangian Multipliers for Mixed Integer Linear Programs

## Abstract

Lagrangian relaxation stands among the most efficient approaches for solving a Mixed Integer Linear Programs (MILP) with difficult constraints. Given any duals for these constraints, called Lagrangian Multipliers (LMs), it returns a bound on the optimal value of the MILP, and Lagrangian methods seek the LMs giving the best such bound. But these methods generally rely on iterative algorithms resembling gradient descent to maximize the concave piecewise linear dual function: the computational burden grows quickly with the number of relaxed constraints.

We introduce a deep learning approach that bypasses the descent, effectively amortizing the local, per instance, optimization. A probabilistic encoder based on a graph convolutional network computes high-dimensional representations of relaxed constraints in MILP instances. A decoder then turns these representations into LMs. We train the encoder and decoder jointly by directly optimizing the bound obtained from the predicted multipliers. Numerical experiments show that our approach closes up to 85 % of the gap between the continuous relaxation and the best Lagrangian bound, and provides a high quality warm-start for descent based Lagrangian methods.

## 1 Introduction

Mixed Integer Linear Programs (MILPs) (Wolsey, 2021) have two main strengths that make them ubiquitous in combinatorial optimization (Korte & Vygen, 2012). First, they can model a wide variety of combinatorial optimization problems. Second, extremely efficient solvers can now handle MILPs with millions of constraints and variables. They therefore have a wide variety of applications. MILP algorithms are exact: they return an optimal solution, or an optimality gap between the returned solution and an optimal one.

MILPs are sometimes hard to solve due to a collection of difficult constraints. Typically, a small number of constraints may link otherwise independent subproblems. For instance, in vehicle routing problems (Golden et al., 2008), there is one independent problem for each vehicle, except for the linking constraints that ensure that exactly one vehicle operates each task of interest. Lagrangian relaxation approaches are popular in such settings as they enable to decouple the different subproblems.

More formally (Conforti et al., 2014, Chap. 8), let $(P)$ be a MILP of the form:

$$(P) \qquad v_P = \min_{\boldsymbol{x}} \boldsymbol{w}^\top \boldsymbol{x}$$

$$s.t.\ \boldsymbol{A}\boldsymbol{x} = \boldsymbol{b} \tag{1a}$$

$$\boldsymbol{C}\boldsymbol{x} = \boldsymbol{d} \tag{1b}$$

$$\boldsymbol{x} \in \mathbb{R}^m \times \mathbb{N}^n \tag{1c}$$

The relaxed Lagrangian problem obtained by dualizing (the difficult) constraints (1a) and penalizing their violation with Lagrangian multipliers (LMs) $\boldsymbol{\pi}$ is:

$$(LR(\boldsymbol{\pi})) \qquad \mathcal{G}(\boldsymbol{\pi}) = \min_{\boldsymbol{x}} \boldsymbol{w}^\top \boldsymbol{x} + \boldsymbol{\pi}^\top (\boldsymbol{b} - \boldsymbol{A}\boldsymbol{x})$$

$$s.t.\ \boldsymbol{C}\boldsymbol{x} = \boldsymbol{d} \tag{2}$$

$$\boldsymbol{x} \in \mathbb{R}^m \times \mathbb{N}^n$$

Standard weak Lagrangian duality ensures that $\mathcal{G}(\boldsymbol{\pi})$ is a lower bound on $v_P$. The Lagrangian dual problem aims at finding the best such bound.

$$(LD) \qquad v_D = \max_{\boldsymbol{\pi}} LR(\boldsymbol{\pi}). \tag{3}$$

Geoffrion theorem (Geoffrion, 1974) ensures that $v_D$ is a lower bound at least as good as the continuous relaxation. It is strictly better on most applications. Beyond this bounds, Lagrangian approaches are also useful to find good solutions of the primal. Indeed, Lagrangian heuristics exploit the dual solution $\boldsymbol{\pi}$ and the primal (possibly infeasible) solution of the relaxed Lagrangian problem $LR(\boldsymbol{\pi})$ to compute good quality solutions of (1). Remark that both the bound and the heuristic work as soon as we have "good" but not necessarily optimal duals $\boldsymbol{\pi}$. By good, we mean Lagrangian duals $\boldsymbol{\pi}$ that lead to a bound $\mathcal{G}(\boldsymbol{\pi})$ which is better than the continuous relaxation, and not too far from $v_D$.

Since $\boldsymbol{\pi} \mapsto \mathcal{G}(\boldsymbol{\pi})$ is piecewise linear and concave, it is generally optimized using a subgradient algorithm. Unfortunately, the number of iterations required to obtain good duals quickly increases with the dimension of $\boldsymbol{\pi}$, which makes the approach extremely intensive computationally.

In this paper, we introduce an encoder-decoder neural network that computes "good" duals $\pi$. The neural network uses state-of-the-art encoder-decoder architecture. The probabilistic encoder $q_{\boldsymbol{\phi}}(\boldsymbol{z}|\iota)$ takes in input $\iota$ a MILP instance as well as the primal and dual solutions of its continuous relaxation, and returns an embedding of the instance as a labelled graph. As is classical when using learning algorithms, we represent a MILP instance as a graph whose vertices are the variables and constraints, and whose edges are the non-zero coefficient of the constraint matrix. The vector representation of each vertex lives in a high dimensional space. The deterministic decoder $f_{\boldsymbol{\theta}}(\boldsymbol{\pi}|\boldsymbol{z})$ projects reconstructs single dimensional duals from constraints labels. We show that the Lagrangian dual function $\mathcal{G}(\boldsymbol{\pi})$ provides a natural loss function. Numerical experiments on two problems from the literature show that the predicted duals close three fourth of the gap between the continuous relaxation and the Lagrangian dual bound. These results are even improved when we make use of the probabilistic nature of the encoder to sample several duals. Finally, when the optimal duals are the target, we show that the predicted duals provide an excellent warm-start for state-of-the-art algorithms for (3).

## 2 LEARNING FRAMEWORK

### 2.1 OVERALL ARCHITECTURE

Iterative algorithms for setting LMs to optimality such as the subgradient method or the Bundle method (BM) start by setting the initial values for LMs. They can be initialized to zero but a solution considered as better in practice by the Combinatorial Optimization community is to take advantage of the CR bound, often cheap to compute. Specifically, optimal values of the CR dual variables identified with the constraints dualized in the Lagrangian relaxation can be understood as LMs. In many problems of interest these LMs are not optimal and can be improved by the subgradient method or BM. We leverage this observation by trying to predict a deviation from the LMs obtained by reinterpreting the continuous relaxation dual solution as a Lagrangian bound.

The architecture is depicted on Figure 1. We start from an input instance $\iota$ of MILP $(P)$ with a set of constraints for which the Lagrangian relaxed problem is easy to compute, we solve $(CR)$ and obtain the corresponding primal and dual solutions. This input is then passed through a probabilistic encoder, composed of three parts: *(i)* the input is represented by a bipartite graph in a similar fashion as in (Gasse et al., 2019) and initial features for the graph nodes are extracted, *(ii)* this graph is passed through a graph neural network in charge of refining the node features by taking into account the structure of the MILP, *(iii)* based on the last layer of the graph neural network, we parametrize a distribution from which we are able to sample a vector $\boldsymbol{z}_c$ for each dualized constraint $c$.

The decoder then translates $\boldsymbol{z}_c$ to a LM $\pi_c = \lambda_c + \delta_c$ by predicting a deviation $\delta_c$ from the CR dual solution variable $\lambda_c$. Finally, the predicted LMs can be used in several ways, in particular to compute a Lagrangian bound or to warmstart an iterative solver.

### 2.2 OBJECTIVE

We train the network's parameters in an end-to-end fashion by maximizing the average Lagrangian dual bound $LR$ defined in (2), obtained from the predicted LMs over a training set. This can be

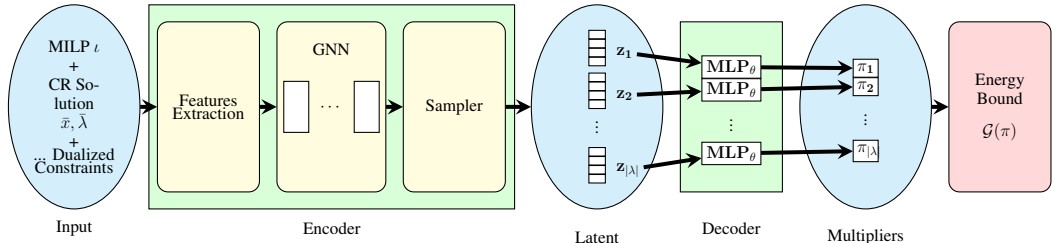

Figure 1: Overall Architecture. From the bipartite graph representation of a MILP and its CR solution, the model computes a Lagrangian dual solution. First the MILP is encoded by a GNN, from which we parametrize a sampler for constraint representations. These representations are then passed through a decoder to compute Lagrangian Multipliers.

cast as an empirical risk optimization, or an Energy-Based Model (Le Cun et al., 2006) with latent variables, where the Lagrangian bound is the (negative) energy corresponding to the coupling of the instance with the subproblem solutions, and the LMs — or more precisely their high-dimensional representations — the latent variables. For our problem, a natural measure of the quality of the prediction is provided by the value of the Lagrangian bound that we want to maximize to tighten the duality gap. Given an instance $\iota$ we want to learn how to predict the latent representations of the LMs from which the Lagrangian bound is the highest:

$$\max_{\phi, \theta} \mathbb{E}_{z \sim q_\phi(\cdot|\iota)} \left[ LR(\lambda + f_\theta(z); \iota) \right]$$

where $q_\phi$ is the probabilistic encoder, mapping each dualized constraint $c$ in $\iota$ to a latent vector $z_c$ computed by independent gaussian distributions, $f_\theta$ is the decoder mapping each $z_c$ to the corresponding LM deviation $\delta_c$ from the CR dual value $\lambda_c$, and $LR$ is the Lagrangian bound.[1] We can observe that this objective has the following properties amenable to gradient-based learning:

1. $LR$ is bounded from above: optimal LMs $\pi^*$ maximize $LR(\cdot)$ over all possible LMs, that is $LR(\pi^*) \geq LR(\pi)$ for any $\pi = \lambda + f_\theta(z)$ Moreover, $LR$ is a concave piece-wise linear function, in other words all optimal solutions will give the same bound.

2. It is straightforward to compute a subgradient w.r.t. to parameters $\theta$. We have that:

$$\nabla_\theta LR(\lambda + f_\theta(z); \iota) = \left( \frac{\partial f_\theta(z)}{\partial \theta} \right)^\top \nabla_\pi LR(\pi; \iota)$$

The jacobian of $f_\theta$ is computed via backpropagation, while $LR$ is simple enough for a subgradient to be given analytically. Provided that $\bar{x}$ is an optimal solution of $LR(\pi)$, we derive:

$$\nabla_{\bar{\pi}} LR(\pi; \iota) = b - A\bar{x}$$

3. For parameters $\phi$, we again leverage function composition and the fact that $q_\phi$ is a gaussian distribution, so we can approximate the expectation by sampling and use the reparametrization trick (Kingma & Welling, 2014; Schulman et al., 2015) to perform standard backpropagation. We implement $q_\phi$ as a neural network, described in details in the following section, returning a mean vector and a variance vector for each dualized constraint $c$, from which a sampler returns a representation vector $z_c$. For numerical stability, the variance is clipped to a safe interval following (Rybkin et al., 2021).

## 2.3 ENCODING AND DECODING INSTANCES

**Encoder** One of the challenges in Machine Learning applications to Combinatorial Optimization is that instances have different input sizes, and so the encoder must be able to cope with these variations to produce high-quality features. Of course this is also the case in many other applications, for instance NLP where texts may differ in size, but there is no general consensus as to what a good

---

[1]With a slight abuse of notation, we use function $f : \mathbb{R}^m \to \mathbb{R}^n$ on *batches* to become $\mathbb{R}^{m \times p} \to \mathbb{R}^{n \times p}$.

feature extractor for MILP instances looks like, contrarily to other domains where variants of RNNs or Transformers have become the de facto standard encoders.

We depart from previous approaches to Lagrangian prediction Sugishita et al. (2021) restricted to instances of the same size, and follow more generic approaches to MILP encoding such as (Gasse et al., 2019; Nair et al., 2020; Khalil et al., 2017) where each instance is converted into a bipartite graph and further encoded by GCNs to compute meaningful feature vectors associated with dualized constraints. Each MILP is converted to a bipartite graph composed of one node for each variable and one node for each constraint. There is an arc between a variable node $n_v$ and a constraint node $n_c$ if and only if $v$ appears in $c$. We differ from Gasse et al. (2019) who add to each arc $(n_v, n_c)$ a weight equal to the coefficient of variable $v$ in constraint $c$. We found these coefficients not useful, on the two datasets that we experimented with during preliminary testing, and thus omitted them.

Each node (variable or constraint) is represented by an initial feature vector $e_n$. We use features similar to (Gasse et al., 2019), see Appendix C for more details. Following (Nair et al., 2020), variables and constraints are encoded as the concatenation of variables features followed by constraint features, of which only one is non-zero, depending on the type of nodes.

To design our stack of GCNs, we take inspiration from structured prediction models for images and texts, where Transformers (Vaswani et al., 2017) are ubiquitous. However, since our input has a bipartite graph structure, we replace the multihead self-attention layers with simple linear graph convolutions[2] (Kipf & Welling, 2016). Closer to our work, we follow Nair et al. (2020) which showed that residual connections (He et al., 2016), dropout (Srivastava et al., 2014) and layer normalization (Ba et al., 2016) are important for the successful implementation of feature extractors for MILP bipartite graphs.

Before the actual GCNs, initial feature vectors $\{e_n\}_n$ are passed through a MLP $F$ to find feature combinations and extend node representations to high-dimensional spaces: $h_n = F(e_n), \forall n$. Then interactions between nodes are taken into account by passing vectors through blocks, represented in Figure 2, consisting of two sublayers.

- The first sublayer connects its input via a residual connection to a layer normalization $LN$ followed by a linear graph convolution $CONV$ of length 1, followed by a dropout regularization $DO$:
$$h'_n = h_n + DO(CONV(LN(h_n)))$$

  The graph convolution passes messages between nodes. In our context, it passes information from variables to constraints, and vice versa.

- The second sublayer takes as input the result of first one, and connects it with a residual connection to a sequence made of a layer normalization $LN$, a MLP transformation and a dropout regularization $DO$:
$$h_n = h'_n + DO(MLP(LN(h'_n)))$$

  This MLP is in charge of finding non-linear interactions between the information collected in the previous sublayer.

This block structure, depicted in Figure 2, is repeated several times, typically 5 times in our experiments, in order to extend the domain of locality. The learnable parameters of a block are the parameters of the convolution in the first sublayer and the parameters of the MLP in the second one. Remark that we start each sublayer with normalization, as it has become the standard approach in Transformer recently Chen et al. (2018). We note in passing that this has also been experimented with by (Gasse et al., 2019) in the context of MILP, although only once before the GCN input, whereas we normalize twice per block, at each block.

Finally, we retrieve the final vectors associated with dualized constraints $\{h_c\}_c$. Each vector $h_c$ is interpreted as the concatenation of two vectors $[z_\mu; z_\sigma]$ from which we compute $z_c = z_\mu + \exp(z_\sigma) \cdot \epsilon$ where elements of $\epsilon$ are sampled from the normal distribution. This concludes the implementation of the probabilistic encoder $q_\phi$.

---

[2]Alternatively, this can be seen as a masked attention, where the mask is derived from the input graph adjacency.

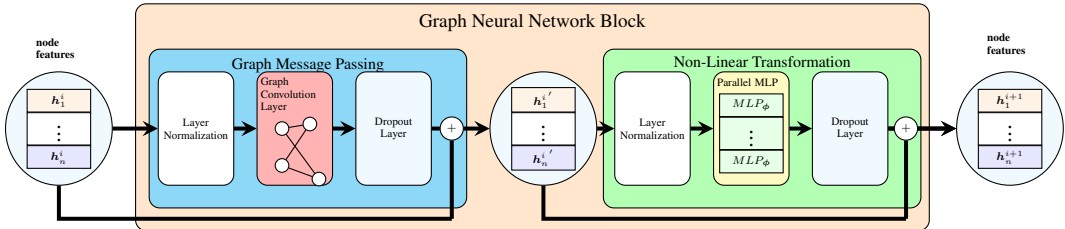

Figure 2: The Graph Neural Network block. The first part is graph message-passing: we apply layer normalization to node features, then convolution over the instance's bipartite graph representation and finally dropout. The second phase consists of normalization, a Multi-Layer perceptron in parallel over all the nodes of the bipartite graph, then dropout. Both sublayers use residual connection between input and output. We apply this block several times to improve feature representations.

**Decoder and Lagrangian Subproblem**   Recall that, in our architecture, from each latent vector representation $z_c$ of dualized constraint $c$ we want to compute the scalar deviation $\delta_c$ to the CR dual value $\lambda_c$ so that the sum of the two improves the Lagrangian bound given by the CR dual solution. In other words, we want to compute $\boldsymbol{\delta}$ such as $\boldsymbol{\pi} = \boldsymbol{\lambda} + \boldsymbol{\delta}$ gives a *good* Lagrangian bound.

For each independent Lagrangian subproblem we want to find its optimal variable assignment, usually with local combinatory constraints, for its objective reparametrized with $\boldsymbol{\pi}$. This approach is typical of structured prediction: we leverage neural networks to extract features in order to compute local energies (scalars), which are utilized by a combinatorial algorithm outputting a structure whose objective value can be interpreted as a global energy. For instance, this is reminiscent of how graph-based syntactic parsing models in NLP compute parse scores (global energies) as sums of arc scores (local energies) computed by RNNs followed by MLPs, where the choice of arcs is guided by well-formedness constraints enforced by a maximum spanning tree solver, see for instance (Kiperwasser & Goldberg, 2016). Thus, the decoder is local to each dualized constraint, and we leverage subproblems to interconnect predictions:

1. We compute LMs (local energies) $\pi_c = \lambda_c + f_{\boldsymbol{\theta}}(z_c)$ for all dualized constraints $c$, where $f_{\boldsymbol{\theta}}$ is implemented as a feed-forward network computing the deviation.

2. For parameter learning or if the subproblems or the Lagrangian bound are the desired output, vector $\boldsymbol{\pi}$ is then passed to the Lagrangian subproblems[3] which compute independently and in parallel their local solutions $x$ and the corresponding values that are summed to give (global energy) $LR(\boldsymbol{\pi})$. The exact computation of $LR$ is of combinatorial nature, problem specific, and is described in Appendix A and in Appendix B.

## 3   RELATED WORK

In this work, we define a Machine Learning model to predict a dual bound for MILP instances sharing common features, which can in turn be used to improve solvers. There is a growing interest in leveraging ML alongside optimization algorithms (Bengio et al., 2018), and particularly with aim of improving MILP solvers (Zhang et al., 2023). Indeed, even though MILP solvers solve problems in an exact way, they make a lot of heuristic decisions for which ML can be used to base these decisions on non-trivial patterns. For instance, classifiers have been designed for Branch and Bound (B&B) algorithms (Lodi & Zarpellon, 2017) in order to choose which variables to branch on (Alvarez et al., 2016; Khalil et al., 2016; Etheve et al., 2020), which B&B node to process (Yilmaz & Yorke-Smith, 2021; Labassi et al., 2022), to decide when to perform heuristics (Hottung et al., 2017; Khalil et al., 2017) or how to schedule them (Chmiela et al., 2021).

More closely related to this contribution, several works have tackled the prediction of high quality primal and dual bounds. For instance, Nair et al. (2020) predict optimal values for large subsets of

---

[3]We use the plural form, but it might be the case that there is only one such problem, depending on the type of problem or instance.

variables, resulting in small MILPs that can be solved to optimality. Another way to provide good primal solutions is to learn to transform an input MILP into an easier problem, solve it and apply a postoptimization procedure to recover primal feasibility of the computed solution (Parmentier, 2023; Dalle et al., 2022). For dual bounds, ML has been employed for cut selection in cutting planes algorithms. Indeed, efficient MILP solvers contain cut generators for strengthening the linear relaxation but a key point is to find a trade off between the improvement of the dual bound and the increasing solving time of the linear relaxation due the added cuts (Dey & Molinaro, 2018). Baltean-Lugojan et al. (2018); Babaki & Jena (2022/09/06); Wang et al. (2023); Balcan et al. (2021); Berthold et al. (2022); Tang et al. (2020); Huang et al. (2022); Afia & Kabbaj (2017) have trained different models for this cut selection. Similarly, Morabit et al. (2021) learn to select good columns at each iteration of a column generation.

Regarding prediction for Lagrangian dual solutions, Nair et al. (2018) consider specifically 2-stage stochastic MILPs, approached by a Lagrangian decomposition for which they learn to predict LMs that would comply with any second-stage scenario to give a good bound on average. Abbas & Swoboda (2022a) propose a new approach to solve linear optimization problems based on Binary Decision Diagrams (Lange & Swoboda, 2021) and a Lagrangian decomposition by constraint. This algorithm can be run on GPU and is trainable. In (Abbas & Swoboda, 2022b), they improve their algorithm by learning an initial dual solution as well as the step in their subgradient method. Other works predict Lagrangian dual solutions for deterministic MILPs like us but stemming from specific combinatorial otpimization problems in contrast to our generic method. Kraul et al. (2023) consider the cutting stock problem and propose a MLP to predict the dual Lagrangian value for each constraint (*i.e.* stock) separately[4]. They use the Lagrangian dual solution to stabilize a column generation using du Merle stabilization method (du Merle et al., 1999). Sugishita et al. (2021) predict a Lagrangian dual solution for the unit commitment problem. In their context, the same problem is solved daily but with different demand forecasts. The prediction is made by a MLP or a random forest and the dual solution is used to warmstart a proximal bundle method, similarly to our evaluation.

In our work, we assume that the decomposition in subproblems is given, but the prediction of good decompositions is also an active avenue of research, to find a good compromise between the quality of the Lagrangian dual bound and the complexity of the relaxed Lagrangian problem. Kruber et al. (2017) train a ML model to determine whether to apply a decomposition and which constraints to dualize while Basso et al. (2020) use ML to classify decompositions from features of the MIP instances.

## 4 EVALUATION

We evaluate our approach on the Multi-Commodity Fixed-Charge Network Design problem and on the Capacitated Facility Location problem. This section describes the dataset considered for the two problems and the numerical results for a series of experiments.

### 4.1 BENCHMARKS

**Multi-Commodity Fixed-Charge Network Design (MCDN)** Given a network with arc capacities and a set of commodities, MCND consists in activating a subset of arcs and routing each commodity from its origin to its destination, possibly fractionned on several paths, using only the activated arcs. The objective is to minimize the total cost induced by the activation of the arcs and the routing of the commodities. This problem is NP-hard and the continuous relaxation provides poor bounds when arc capacities are high. Hence, it is usually tackled with Lagrangian relaxation-based methods (Akhavan Kazemzadeh et al., 2022). We describe the mathematical formulation as a MILP of this problem, its Lagrangian Relaxation and subproblems in Appendix A.

Since there is no publicly available dataset for this problem adapted to Machine Learning, with large collections of instances sharing common features, we designed four small datasets on which we can run experiments. We generate four MCND datasets from a subset of instances of the Canad dataset (Crainic et al., 2001), well known and commonly used by the optimization community to benchmark solvers. The first two datasets, MCND-SMALL-COM40 and MCND-SMALL-COMVAR, contain instances

---

[4]Another MLP is proposed predicting all dual Lagrangian values at once but it is limited to instances containing no more than a fixed number of stocks.

which all share the same network (20 nodes and 230 edges) and the same arc capacities and fixed costs, but with different values for origins, destinations, volumes and routing costs. Instances of the former all involve the same number of commodities (40), while for the latter the number of commodities can also vary from 40 to 200. Dataset MCND-BIG-COM40 is generated similarly to MCND-SMALL-COM40 but upon a bigger graph containing 30 nodes and 500 arcs. Finally, MCND-BIG-COMVAR contains examples generated using either the network of MCND-SMALL-COM40 or the one of MCND-BIG-COM40, with the number of commodities varying between 40 and 200. More details can be found in Appendix D.

**Capacitated Facility Location (CFL)**   CFL consists, given a set of customers and a set of facilities, in deciding which facilities to open in order to serve the customers at minimum cost, defined as the sum of the fixed costs associated with the opening of the facilities plus the sum of the service costs between facilities and customers.

We generate one dataset CFL. Each example has either 16, 25 and 50 facilities, and 50 customers. For a given number of facilities, the fixed costs and capacities are the same but customer demands and service costs vary.

## 4.2   NUMERICAL RESULTS

We want to evaluate how our Lagrangian bound prediction compares to an iterative model based on subgradient, and how useful the former is as an initial point to warmstart the latter. For that purpose, we choose a state-of-the-art proximal bundle solver provided by SMS++ (Frangioni et al., 2023) which allows writing a MILP in a block structure fashion and using decomposition techniques to solve subproblems efficiently. We also compare our approach with CR computed using CPLEX[5] optimiser.

All MILP instances for which we want to evaluate our model on are first solved by SMS++. For an instance $\iota$ we denote $\boldsymbol{\pi}_\iota^*$ the LMs returned by SMS++, and $\widehat{\boldsymbol{\pi}}_\iota$ the LMs returned by our model. We drop the index when instance is clear from the context. Recall from Section 1 that given an instance $\iota$ and Lagrangian multipliers vector $\boldsymbol{\pi}$ we denote by $LR(\boldsymbol{\pi}; \iota)$ the objective value of the Lagrangian bound. We write $CR(\iota)$ the value returned by the continuous relaxation of $\iota$.

Finally, when evaluating we do not sample constraint representations but rather take the modes of their distributions. In practice, following the notations from Section 2.3 we set $\boldsymbol{z}_c = \boldsymbol{z}_\mu$ for each dualized constraint $c$.

**Metrics**   We want to measure how close our prediction is to the solution returned by BM, considered as a proxy to the optimal solution, and how it compares as a starting point for BM with the all-zeros vector and CR dual solution interpreted as a Lagrangian bound. Hence, we use two metrics which average these measures over a dataset $\mathcal{I}$:

- the mean gap percentage (GAP):

$$100\frac{1}{|\mathcal{I}|} \sum_{\iota \in \mathcal{I}} \frac{LR(\boldsymbol{\pi}^*; \iota) - LR(\widehat{\boldsymbol{\pi}}; \iota)}{LR(\boldsymbol{\pi}^*; \iota)}$$

  GAP measures the optimality of our prediction. The GAP is equal to zero when we predict exactly a vector of optimal Lagrangian multipliers.

- the mean gap closure percentage w.r.t. to continuous relaxation (GAP-CR):

$$100\frac{1}{|\mathcal{I}|} \sum_{\iota \in \mathcal{I}} (1 - \frac{LR(\boldsymbol{\pi}^*; \iota) - LR(\widehat{\boldsymbol{\pi}}; \iota)}{LR(\boldsymbol{\pi}^*; \iota) - CR(\iota)})$$

  GAP-CR measures how our prediction compares to CR. It is negative if the prediction provides a bound worse than the continuous relaxation, and positive if it is better. Moreover, it is equal to 100 if the bound is the same as the optimal Lagrangian bound.

---

[5]https://www.ibm.com/fr-fr/analytics/cplex-optimizer

**Data for Evaluation**   We use cross-validation to evaluate our model. Each dataset is partitioned in 10 subsets, or folds. Each element of the partition is tested with a model trained on its complement, where we divide the complement as 90% train, 10% validation. Results are averaged over folds.

**Bound Accuracy**   In Table 1 we report how our model behaves compared to the optimal Lagrangian bound given by our BM solver, and how it compares with the CR bound. Our model can reach 2% difference with BM on MCND-SMALL-COM40, the easiest corpus with a small fixed network and a fixed number of commodities. This means that one pass through our network can save numerous iterations if we can accept about 2% bound error on average. When the number of commodities also varies, as in MCND-SMALL-COMVAR, we see that our model GAP is twice as high, reaching about 4%. From the GAP-CR results on these two datasets, we can see that we our model can effectively predict a solution different from CR and is able to close almost 85% of the gap between the CR bound and the Lagrangian bound on MCND-SMALL-COM40. On MCND-SMALL-COMVAR the results are analog: when the number of commodities varies, our model is less accurate.

We can see a similar trend on MCND-BIG-COM40, MCND-BIG-COMVAR and CFL with results slightly lower. This might be due to the fact that the datasets are more difficult, or simply because we explored hyper-parameters on MCND small datasets, and it might be the case that they are suboptimal for bigger graphs and different MILPs.

| Dataset | GAP % (↓) | | | GAP-CR % (↑) | | |
|---|---|---|---|---|---|---|
| | train | validation | test | train | validation | test |
| MCND-SMALL-COM40 | 1.75 | 2.09 | 2.01 | 85.82 | 84.44 | 84.99 |
| MCND-SMALL-COMVAR | 4.53 | 4.22 | 4.02 | 82.09 | 81.66 | 82.39 |
| MCND-BIG-COM40 | 3.45 | 3.61 | 3.70 | 76.37 | 76.11 | 75.84 |
| MCND-BIG-COMVAR | 5.01 | 4.58 | 4.52 | 78.06 | 78.38 | 78.38 |
| CFL | 16.57 | 16.87 | 16.93 | 46.84 | 47.65 | 48.02 |

Table 1: Evaluation results for both metrics are averaged over the different folds of the dataset.

**Bundle Method Warmstart**   In Table 2 we compare different initial Lagrangian Multipliers vectors on the validation set of MCND-BIG-COMVAR. We run our Bundle Method solver until the threshold $\epsilon$ is greater than the difference between $LR(\pi^*)$ and the current Lagrangian bound. We average resolution time and number of iterations over instance, and compute standard deviation. Three initialization methods are compared: initializing LMs with zero, using the CR dual solution, and the prediction of our model.

We can see that CR is really not competitive with the null vector initialization, since the the small gain in number of iterations is absorbed by the supplementary computation. On the other hand, our method based on prediction shows a significant improvement over the other two initialization methods. Resolution time is more roughly halved for the coarse threshold, and above one third faster for the fine one. This is expected, as gradient based method naturally slow down as they approach convergence.

| $\epsilon$ | zero | | CR | | learning | |
|---|---|---|---|---|---|---|
| | time (s) | # iter. | time (s) | # iter. | time (s) | # iter. |
| $10^{-1}$ | $40.31_{\pm 88.82}$ | $133.56_{\pm 90.52}$ | $36.92_{\pm 78.33}$ | $121.43_{\pm 105.91}$ | $\mathbf{18.29}_{\pm 43.54}$ | $\mathbf{81.96}_{\pm 68.22}$ |
| $10^{-2}$ | $79.78_{\pm 202.45}$ | $213.61_{\pm 212.39}$ | $75.58_{\pm 199.38}$ | $207.08_{\pm 224.26}$ | $\mathbf{47.32}_{\pm 125.81}$ | $\mathbf{148.96}_{\pm 153.21}$ |
| $10^{-3}$ | $113.22_{\pm 307.17}$ | $276.89_{\pm 306.81}$ | $109.46_{\pm 284.97}$ | $271.94_{\pm 333.04}$ | $\mathbf{76.92}_{\pm 215.47}$ | $\mathbf{198.88}_{\pm 213.18}$ |

Table 2: Comparisons of LM initialization for a Bundle solver on the first two folds of MCND-BIG-COMVAR. We consider the all-zeros vector initialization, the continuous relaxation duals, and the prediction of our model. For each, we give mean time and standard deviation.

**Ablation Study** In Table 3 we compare four variants of our original model, denoted `full`, on the first fold of MCND-BIG-COMVAR. In the first variant `-sum`, the dual solution values are passed as constraint node features but are not added to output of the decoder to produce LMs. The network must transport these values from its input layer to its output. In the second variant, `-cr` the CR solution is not given as input features to the network. This is challenging because because the network does not have access to a good starting point. The last variant, `-sample_latent`, use CR as `full` but does not sample representations $z_c$ in the latent domain during training, but rather sample a LM deviation $\delta_c$ directly.

We can see that performance of `-sum` are on par with `full` for the GAP metrics, an a little lower for GAP-CR, while `-cr` cannot return acceptable bounds. This indicates that the CR solution passed as input features is essential for our architecture, whereas the computation of the deviation instead the full LM directly is not an important trait. However, we found that training showed more stability when the objective was to predict deviation, see Appendix F for more details.

Learning with sampling in the latent space (`full`) is slightly better than learning with sampling deviations directly (`-sample-latent`), and performance are also more stable along the training process (cf. Appendix F).

|  | GAP % ($\downarrow$) | GAP-CR % ($\uparrow$) |
|---|---|---|
| `full` | 4.52 | 78.64 |
| `-sum` | 4.52 | 76.51 |
| `-cr` | 14.87 | 14.55 |
| `-sample_latent` | 4.59 | 77.29 |

Table 3: Ablation study comparing the prediction of LMs rather than deviation (`-sum`), not using CR features `-cr`, or learning with sampling in the output domain `-sample-latent`.

## 5 CONCLUSION

We have presented a novel method to compute good Lagrangian dual solutions for MILPs sharing common attributes, by predicting Lagrangian multipliers. We cast this problem as an encoder-decoder prediction, where the probabilistic encoder outputs one distribution per dualized constraint from which we can sample constraint vecetor representation. Then a decoder transforms these representation into Lagrangian multipliers.

We showed experimentally that this method gives bounds significatively better that the commonly used heuristics, *e.g.* it is able to reduce the gap to the optimal bound by $85\%$, and that when used to warmstart an iterative solver tom compute the optimal Lagrangian lower bound, the predicted point can reduce the solving times by a large margin.

Our predictions could be exploited in primal heuristics, possibly with auxiliary losses predicting values from variable nodes, or to efficiently guide a Branch-and-Bound exact search.

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

## A  MULTI COMMODITY CAPACITATED NETWORK DESIGN PROBLEM

A MCND instance is given by a directed simple graph $D = (N, A)$, a set of commodities $K$, an arc-capacity vector $c$, and two cost vectors $r$ and $f$. Each commodity $k \in K$ corresponds to a triplet $(o^k, d^k, q^k)$ where $o^k \in N$ and $d^k \in N$ are the nodes corresponding to the origin and the destination of commodity $k$, and $q^k \in \mathbb{N}^*$ is its volume. For each arc, $(i, j) \in A$, $c_{ij} > 0$ corresponds to the maximum amount of flow that can be routed through $(i, j)$ and $f_{ij} > 0$ corresponds to the fixed cost of using arc $(i, j)$ to route commodities. For each arc $(i, j) \in A$ and each commodity $k \in K$, $r_{ij}^k > 0$ corresponds to the cost of routing one unit of commodity $k$ through arc $(i, j)$.

A MCND solution consists of an arc subset $A' \subseteq A$ and, for each commodity $k \in K$, in a flow of value $q^k$ from its origin $o^k$ to its destination $d^k$ with the following requirements: all commodities are only routed through arcs of $A'$, and the total amount of flow routed through each arc $(i, j) \in A'$ does not exceed its capacity $c_{ij}$. The solution cost is the sum of the fixed costs over the arcs of $A'$ plus the routing cost, the latter being the sum over all arcs $(i, j) \in A$ and all commodities $k \in K$ of the unitary routing cost $r_{ij}^k$ multiplied by the amount of flow of $k$ routed through $(i, j)$.

### A.1  MILP FORMULATION

A standard model for the MCND problem (Gendron et al., 1999) introduces two sets of variables: the continuous flow variables $x_{ij}^k$ representing the amount of commodity $k$ that is routed through arc $(i, j)$ and the binary design variables $y_{ij}$ representing whether or not arc $(i, j)$ is used to route commodities. Denoting respectively by $N_i^+ = \{j \in N \mid (i, j) \in A\}$ and $N_i^- = \{j \in N \mid (j, i) \in A\}$ the sets of forward and backward neighbors of a vertex $i \in N$, the MCND problem can be

modeled by the following mixed binary linear problem:

$$\min \sum_{(i,j) \in A} \left( f_{ij} y_{ij} + \sum_{k \in K} r_{ij}^k x_{ij}^k \right) \tag{4}$$

$$s.t. \sum_{j \in N_i^+} x_{ij}^k - \sum_{j \in N_i^-} x_{ji}^k = b_i^k \qquad \forall i \in N, \forall k \in K, \tag{5}$$

$$\sum_{k \in K} x_{ij}^k \leq c_{ij} y_{ij}, \qquad \forall (i,j) \in A, \tag{6}$$

$$0 \leq x_{ij}^k \leq q^k \qquad \forall (i,j) \in A, \forall k \in K, \tag{7}$$

$$y_{ij} \in \{0,1\}, \qquad \forall (i,j) \in A, \tag{8}$$

where

$$b_i^k = \begin{cases} q^k & \text{if } i = o^k, \\ -q^k & \text{if } i = d^k, \\ 0 & \text{otherwise.} \end{cases}$$

The objective function (4) minimizes the sum of the routing and fixed costs. Equations (5) are the flow conservation constraints that properly define the flow of each commodity through the graph. Constraints (6) are the capacity constraints ensuring that the total amount of flow routed through each arc does not exceed its capacity or is zero if the arc is not used to route commodities. Finally inequalities (7) are the bounds for the $x$ variables and inequalities (8) are the integer constraints for the design variables.

The flow of each commodity can be restricted, without loss of generality, to not entering its origin node nor leaving its destination one. This can be done by adding the following equations to the model:

$$x_{ij}^k = 0 \text{ for each commodity } k \in K \text{ and for each arc } (i,j) \in A \text{ s.t. } i = d^k \text{ or } j = o^k \tag{9}$$

## A.2 LAGRANGIAN KNAPSACK RELAXATION

A standard way to obtain good bounds for the MCND problem is to solve the Lagrangian relaxation obtained by dualizing the flow conservation constraints (5) in formulation (4)-(9). Let $\pi_i^k$ be the Lagrangian multiplier associated with node $i \in N$ and commodity $k \in K$. Dualizing the flow conservation constraints gives the following Lagrangian problem:

$$(LR(\pi)) \quad \min_{(x,y) \text{ satisfies } (6)-(9)} \sum_{(i,j) \in A} \left( f_{ij} y_{ij} + \sum_{k \in K} r_{ij}^k x_{ij}^k \right) + \sum_{k \in K} \sum_{i \in N} \pi_i^k \left( b_i^k - \sum_{j \in N_i^+} x_{ij}^k + \sum_{j \in N_i^-} x_{ji}^k \right)$$

Rearranging the terms in the objective function and observing that the Lagrangian relaxation decomposes by arcs, we obtain a sub-problem for each arc $(i,j) \in A$ of the form:

$$(LR_{ij}(\pi)) \quad \min f_{ij} y_{ij} + \sum_{k \in K_{ij}} (r_{ij}^k - \pi_i^k + \pi_j^k) x_{ij}^k \tag{10}$$

$$s.t. \sum_{k \in K_{ij}} x_{ij}^k \leq c_{ij} y_{ij}, \tag{11}$$

$$0 \leq x_{ij}^k \leq q^k, \qquad \forall k \in K_{ij}, \tag{12}$$

$$y_{ij} \in \{0,1\}, \tag{13}$$

where, $K_{ij} = \{k \in K \mid j \neq o^k \text{ and } i \neq d^k\}$ is the set of commodities that may be routed through arc $(i,j)$.

For each $(i,j) \in A$, $LR_{ij}(\pi)$ is a MILP with only one binary variable. If $y_{ij} = 0$, then, by (11) and (12), $x_{ij}^k = 0$ for all $k \in K_{ij}$. If $y_{ij}$, the problem reduces to a continuous knapsack problem. An optimal solution is obtained by ordering the commodities of $K_{ij}$ with respect to decreasing values

$r_{ij}^k - \pi_i^k + \pi_j^k$ and setting for each variable $x_{ij}^k$ the value $\max\{\min\{q^k, c_{ij} - \sum_{k \in K(k)} q^k\}, 0\}$ where $K(k)$ denotes the set of commodities that preceed $k$ in the order. This step can be done in $O(|K_{ij}|)$ if one computes $x_{ij}^k$ following the computed order. Hence, the complexity of the continuous knapsack problem is $O(|K_{ij}| \log(|K_{ij}|))$. The solution of $LR_{ij}(\pi)$ is the minimum between the cost of the continuous knapsack problem and $\mathbf{0}$.

Lagrangian duality implies that the value $LR(\pi) = \sum_{(i,j) \in A} LR_{ij}(\pi) + \sum_{i \in N} \sum_{k \in K} \pi_i^k b_i^k$ is a lower bound for the MCND problem and the best one is obtained by solving the following Lagrangian dual problem:

$$(LD) \quad \max_{\pi \in \mathbb{R}^{N \times K}} LR(\pi)$$

## B   Capacitated Facility Location Problem

A CFL instance is defined by a set $J$ of facilities and a set $K$ of customers. With each facility $j \in J$ is associated a capacity $c_j$ and a fixed cost $f_j$. A demand $q^k$ is associated with each customer $k \in K$. Finally, a service cost $r_j^k$ is associated with each facility $j \in J$ and each customer $k \in K$ and corresponds to the cost of serving one unit of demand of customer $k$ from facility $j$.

A CFL solution consists in a subset of open facilities as well as the amount of demand served from these open facilities to each customer. Its cost is the sum of the fixed costs over the open facilities plus the sum over every facility $j \in J$ and every customer $k \in K$ of the unitary service cost $r_j^k$ multiplied by the amount of demand of customer $k$ served from facility $j$.

### B.0.1   MILP formulation

A standard model for the CFL (Akinc & Khumawala, 1977) introduces two sets of variables: the continuous variables $x_j^k$ representing the amount of demand of customer $k$ served from facility $j$, and the binary variables $y_j$ indicating whether facility $j \in J$ is open. Hence, a formulation of the problem is:

$$\min \sum_{j \in J} \left( f_j y_j + \sum_{k \in K} r_j^k x_j^k \right) \tag{14}$$

$$s.t. \sum_{j \in J} x_j^k = q^k \qquad \forall k \in K \tag{15}$$

$$\sum_{k \in K} x_j^k \leq c_j y_j \qquad \forall j \in J \tag{16}$$

$$0 \leq x_j^k \leq q^k \qquad \forall j \in J, \ \forall k \in K \tag{17}$$

$$y_j \in \{0, 1\} \qquad \forall j \in J \tag{18}$$

The objective function (14) minimizes the total cost. Equations (15) impose that the demand is every customer is served. Constraints (16) are the capacity constraints that ensure that the amount of demand served by a facility $j \in J$ does not exceed its capacity $c_j$, or is zero if the facility is not open. Constraints (17) are the bounds constraints on the $x$ variables and constraints (18) force the variables $y$ to be integer.

### B.0.2   Lagrangian Relaxation

A Lagrangian relaxation of the CFL problem is obtained by dualizing (15). In that case, the Lagrangian dual gives a better bound than the continuous relaxation (Geoffrion, 1974) while the relaxed problem can be solved efficiently as it decomposes by facility. Let $\pi^k$ be the Lagrangian multiplier of constraint (15) associated with customer $k \in K$. The relaxed problem $LR(\pi)$ is given by:

$$(LR(\pi)) \quad \min_{(x,y) \text{ satisfies } (16)-(18)} \sum_{j \in J} f_j y_j + \sum_{k \in K} q^k \pi^k + \sum_{j \in J} \sum_{k \in K} (r_j^k - \pi^k) x_j^k$$

The relaxed problem $LR(\pi)$ can be decomposed by facility and the sub-problem associated with facility $j \in J$ is the following:

$$(LR_j(\pi)) \quad \min f_j y_j + \sum_{k \in K} (r_j^k - \pi^k) x_j^k \tag{19}$$

$$s.t. \sum_{k \in K} x_j^k \le c_j y_j \tag{20}$$

$$0 \le x_j^k \le q^k \qquad \forall k \in K \tag{21}$$

$$y_j \in \{0, 1\} \tag{22}$$

The value of the relaxed Lagrangian problem $LR(\pi)$ is equal to $\sum_{j \in J} LR_j(\pi) + \sum_{k \in K} q^k \pi^k$. $LR(\pi)$ contains only one non continuous variable. If $y_j = 0$ equals 0, then $x_j^k = 0$ by constraints (20) and (21). If $y_j = 1$, then the problem reduces to a continuous knapsack problem which can be solved by ordering the customers following decreasing values $r_j^k - \pi^k$ and setting $x_j$ to $\max\{\min\{q^k, c_j - \sum_{k \in K(k)} q^k\}, 0\}$ where $K(k)$ denotes the set of customers that preceed $k$ in the order. Solving $LR_j(\pi)$ consists in choosing among these two solutions the one which is minimum so $LR_j(\pi)$ can be solved in $O(|K| \log(|K|))$. The best Lagrangian lower bound for CFL can be found by solving the Lagrangian dual problem:

$$\max_{\pi \in \mathbb{R}^K} LR(\pi)$$

## C  INITIAL FEATURES

In order to extract useful features, we define a network based on graph convolutions presented in Figure 1, following earlier the work of (Gasse et al., 2019) on MILP encoding. We give the initial node features of the MILP-encoding bipartite graph presented in Section 2.3.

Given an instance of the form:

$$(P) \qquad \min \boldsymbol{w}^\top \boldsymbol{x}$$
$$s.t. \ \boldsymbol{A}\boldsymbol{x} = \boldsymbol{b}$$
$$\boldsymbol{x} \in \mathbb{R}^k \times \mathbb{N}^n$$

with CR primal solution $\bar{\boldsymbol{x}}$ and dual solution $\boldsymbol{\lambda}$, we consider the following initial features for a variable $x_i$:

- the coefficient of the variable in the objective function $w_i$;
- the value of the corresponding variable in the primal solution of the continuous relaxation $\bar{x}_i$;
- the reduced cost of the variable for this value $(\boldsymbol{w} - \boldsymbol{A}^\top \boldsymbol{\lambda})_i$;
- a binary value that states whether $x_i$ is integral or real.

For constraint $c$, we consider:

- the righthand-side of the constraint $b_c$;
- the value of the associated dual variable in the solution of the continuous relaxation of the problem $\lambda_c$;
- one binary value that states whether $c$ is an inequality constraint with $\le$ and another for $\ge$ (both will be equal to one for equality constraints);
- one binary value states whether $c$ is to be dualized.

## D  DATASET COLLECTION DETAILS

In this appendix we provide further details on the dataset construction.

**Multi-Commodity Fixed-Charge Network Design**    The Canad dataset (Crainic et al., 2001) is composed of instances of different sizes. We consider easier and harder instances, the easier have a Lagrangian Dual that can be solved in nearly one second and the hardest in approximately one hour. For each fixed graph support and number of commodities, 4 different instances differ one from another:

- the dominance of the fixed costs w.r.t. the routing costs,
- the dominance of the capacities w.r.t. the volumes of the commodities.

To create our dataset we select two different graph supports: one smaller with 20 nodes and 230 edges and one bigger with 30 nodes and 500 edges. For the graph with 20 nodes, we consider 40, 80, 120, 160, and 200 commodities. For the graph with 30 nodes, we consider 40, 80, and 120 commodities. For the bigger graph, we do not consider 160 and 200 instances in the dataset as to solve at optimum one instance we need more than 1 hour, but we test one model using some of those instances to test generalizability.

In terms of support graph dimension and number of commodities we consider the following variants:

- MCND-SMALL-COM40 :2000 MCND instance with fixed support graph (20 nodes, 230 edges) and fixed commodity number (40)
- MCND-SMALL-COMVAR : 4600 MCND instance with fixed support graph (20 nodes, 230 edges) and variable commodity number (40,80,120,160,200)
- MCND-BIG-COM40 : 2400 MCND instance with fixed support graph (30 nodes, 500 edges) and fixed commodity number (40)
- MCND-BIG-COMVAR : 3700 MCND instance with fully heterogeneous fixed support graph

For the Multi-Commodity Capacitated Network Design Problem, in all the datasets, we change the routing costs, the node of the origin and the destination of the commodities, and their volume. We keep the capacities and the fixed costs fixed for three reasons:

- As we change the routing costs it is redundant to change the fixed ones and the same holds for volumes and capacities.
- The four instances p33,p34,p35, and p36 already describe different relations between fixed/routing costs and capacities/potentials, and we generate the instances for the datasets MCND-SMALL-COM40 and MCND-SMALL-COMVAR starting from these 4.
- The same holds for the four instances p49,p50,p51, and p52, but these instances have 30 nodes and 500 edges. We generate the instances for the dataset MCND-BIG-COM40 starting from these 4.
- The dataset MCND-BIG-COMVAR is constructed considering both

To create these matrices we randomly generate the vectors considering a normal distribution with mean and variance equal to the ones of the vector for the original instance, taking a vector $v \in \mathbb{R}^d$ of the starting instance, we construct the new vector using $d$ samples of a normal distribution with mean $\mathbb{E}[v]$ and standard deviation $\sqrt{std(v)}$.

**Capacitated Facility Location**    To construct the CFL instances we start from some instances contained in the ORLib (Beasley, 1990). For the CFL problem, we construct only one dataset of 6560 instances. The dataset CFL is composed of 6560 CFL instances with different facilities (16,25,50) with the same number of customers (50). We take instances with In order to ensure the feasibility, we consider a different modification for these vectors. Multiply the old data by one normal variable translate of 0.5 to have all values between 0.5 and 1.5. Then we shuffle that vector in order to exploit symmetries better.

## E    HYPERPARAMETERS

**Model Architecture**    For all datasets, the MLP $F$ from initial features to high-dimensional is implemented as a linear transformation (10 to 250) followed by an activation followed by a linear

tranformation to the size of internal representation of nodes for the GNN, set to 500 for the dataset with 40 commodities and 250 for the ones with different number of commodities.

For MCDN we use 5 blocks, while for CFL we use only 3. Th the hidden layer of the MLP in the second sublayer of each block has size 1000.

The decoder is am MLP with one hidden layer of 125 nodes.

All non-linear activations are implemented as ReLU.

The dropout rate is set to 0.25.

**Optimiser Specifications** We use as optimizer RAdam, with learning rate 0.001, a clip Norm to one and exponential decay with initial learning rate 0.1, decay 0.9, step size 100000 and minimum learning rate $10^{-8}$.

**GPU specifics** For the datasets MCND-SMALL-COM40 and MCND-BIG-COM40 we use GPUs Nvidia Quadro RTX 5000 with 16 GB of RAM. For the datasets MCND-SMALL-COMVAR, MCND-BIG-COMVAR and CWL we use Nvidia A40 GPUs accelerators with 48Gb of RAM.

**CPU specifics** The warm starting of the proximal Bundle in SMS++ needs only CPU, the experiment are done on Intel Core i7-8565U CPU @ 1.80GHz × 8.

## F ABLATION STUDY – COMPLEMENTS

Figure 3 presents the GAP-CR score on the validation set of first fold of MCDN-BIG-COMVAR after each epoch of training for the models considered in the ablation study.

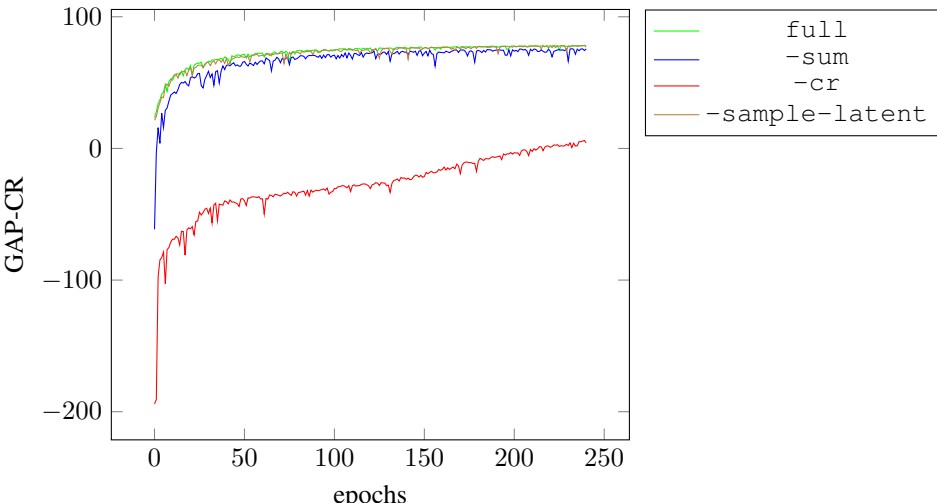

Figure 3: Ablation: comparison of performance over training epochs. In green the `full` model, in blue `-sum`, in red `-cr`, in brown `-sample-latent`, see description in Section 4.2.

