# OpenReview forum: "PREDICTING ACCURATE LAGRANGIAN MULTIPLIERS FOR MIXED INTEGER LINEAR PROGRAMS"
_ICLR.cc/2024/Conference — Submitted to ICLR 2024_

### Official Review · Reviewer_4s9N · 2023-10-23

**Soundness:** 2 fair
**Presentation:** 3 good
**Contribution:** 2 fair
**Rating:** 5
**Confidence:** 4

**Summary:**

Lagrangian decomposition is an approach to obtain lower bounds for optimal values of hard combinatorial optimization problems. For some problems and decompositions, these bounds are tighter than the simple continuous relaxation (which just drops the integrality constraint). The lower bound in Lagrangian decomposition is a concave piecewise-affine function of the Lagrange multipliers and is traditionally maximized using subgradient or bundle methods, which may be slow.

The paper proposes a deep learning architecture to predict optimal values of Lagrange multipliers in Lagrangian decomposition of MILP problems. The motivation is to use these predicted suboptimal LMs to warm-start subgradient or bundle methods.

The architecture is a encoder-decoder one. The probabilistic encoder encodes the input MILP instance and the primal+dual optimal solutions of the continuous relaxation into a latent space. The deterministic decoder then decodes these latent features to the values of Lagr. multipliers (precisely, to differences between the LMs in Lagr. decomposition and continuous relaxation).

The method is tested on two MILP problems: multicommodity fixed-charge network design (MCDN) and capaciated facility location (CFL). These have natural decompositions to small subproblems, which provide strictly better bounds than continuous (LP) relaxations. The predicted LMs are compared to the LMs obtained from continuous relaxations. This shows that the predicted LMs sometimes close 3/4 of the gap between the optimal lower bound and the continuous-relaxation lower bound. Moreover, the runtime of the bundle solver is compared when initialized with (a) zero LMs, (b) LMs from the continuous relaxation, (c) LMs from the proposed method. Warm-starting by the predicted LMs speeds up the bundle solver typically by tens of percents.

**Strengths:**

To my understanding, the method is more general than the previous methods to predict optimal Lagrange multipliers. However, the idea of predicting Lagrange multipliers was proposed before.

The topic itself (predicting optimal Lagrange multipliers in Lagr. decomposition) is relevant for combinatorial optimization. However, in my opinion, its impact is more limited than, e.g., predicting decisions in branch&bound search.

The deep learning architecture is, to my knowledge, novel. However, this novelty is only incremental as the architecture combines known techniques in a novel way.

The text is clear enough, up to inconsistent notation and its frequent abuse.

**Weaknesses:**

First let me admit that I am not an expert in deep learning but I have good knowledge of combinatorial optimization and Lagrangian decompsition. So I will comment mainly on the latter.

The two MILP problems (MCDN, CFL) on which the method is tested have very specific decompositions: the subproblems are small (each sitting on an edge or node of the problem graph) and each subproblem has only one integer (0-1) variable. In particular, both subproblems are almost identical: they are continuous knapsack problems with an additional indicator variable than switches the edge/node on and off. It is possible that the relatively good reported performance would not extend to decompositions to more complex subproblems. Even if the method did not perform well on more complex problems, it would nevertheless be useful to report it. In my opinion, this significantly reduces the impact of the work.

The approach is applicable not only to MILPs but also ILPs or 0-1 LPs. A good source of more complex decompositions is the 0-1 LP formulation of the max-apriori (MAP) inference problem in graphical models (aka discrete energy minimization, aka Weighted Constraint Satisfaction Problem). This problem can be decomposed to arbitrary subproblems, each of which is itself a MAP inference problem. See e.g. [1,2,6,7]. While tree-structured subproblems provide the same bound as the continuous (LP) relaxation, non-tree subproblems (such as cycles or planar graphs [4,5]) provide strictly tighter bounds. There is a large public database of instances, e.g. [3].

Moreover, I wonder if the method is competitive to some other methods to suboptimally compute Lagrange multipliers, not based on learning. One example is min-marginal averaging -- see [Lange2021, Abbas2022a] and references therein. Though this method (without smoothing) is only suboptimal, it is much faster than subgradient methods, especially if the subproblems are small. Let me hypothesize that for MCDN and CFL, a few iterations of min-marginal averaging, warm-started by continuous relaxation, would close a large part of the gap and be faster  than prediction based on deep learning.

[1] J. K. Johnson, D. M. Malioutov, and A. S. Willsky.
Lagrangian relaxation for MAP estimation in graphical models.
Allerton Conf. Communication, Control and Computing, 2007.

[2]  N. Komodakis, N. Paragios, and G. Tziritas.
MRF optimization via dual decomposition: Message-passing revisited.
ICCV 2007.

[3] Kappes et al.
A Comparative Study of Modern Inference Techniques for Discrete Energy Minimization Problems.
IJCV 2015.

[4] Yarkoni, J.
Planar Decompositions and Cycle Constraints.

[5] Batra et al.
Beyond Trees: MAP Inference in MRFs via Outer-Planar Decomposition.

[6] M. Wainwright.
Graphical Models, Exponential Families, and Variational Inference.
2008.

[7] T Werner.
Revisiting the Linear Programming Relaxation Approach to Gibbs Energy Minimization and Weighted Constraint Satisfaction.
PAMI 2010.

Minor comments:
- The word 'accurate' in the title is redundant and misleading. I'd replace it with `optimal'.
- The notation is quite often inconsistent and non well designed. E.g.:
- The decoder is denoted by $f(\pi\mid z)$ in the intro, which is confusing because it is deterministic (it is correct later).
- The symbols $LR(\pi)$ and ${\cal G}(\pi)$ in (2) apparently denote the same thing.
- Section 2.1: The bipartite graph encoding the MILP constraints is known as factor graph.
- Typo below (13): $y_{ij}$ should be $y_{ij}=1$.
- Typo below (18): "demand is ... is"

POST REBUTTAL: I still find the paper not strong enough, mainly for limited instance class in the experiments. Therefore, I keep my evaluation.

**Questions:**

It is rather surprising that the coefficients of the variables in MILP constraints were not needed for training (as noted in the 2nd par of section 2.3). This would not surprise me if the MILP formulations had all coefficients similar (incl. their signs) - but this is not the case (there are $r_{ij}^k,b_i^k,c_{ij}$ in the MILP formulation of MCDN, similarly for CFL). Do you have any insight, please?

Do you plan to make the code available if the paper is accepted?

---

> ### Author Response · Authors · 2023-11-15
>
> Thank you for you valuable review.
> We appreciate that you find the text clear.
> We agree that our method is more general and that the topic is relevant.
> The architecture that we propose is also novel, as you mention, and is a condensed version of the effcient MILP represenation that have been proposed recently.
>
> You are right, predicting Lagrangian multipliers is not new but our approach is more generic since it is not problem specific but it can be applied for computing any Lagrangian Relaxation of a compact mixed integer linear problem.
>
> The impact of predicting LM is maybe more limited than predicting the variable to branch on or the node to process. However, branch-and-bounds methods heavily depend on primal and dual bounds for which our method is helpful: it provies dual bounds and can be used to derive primal bounds using Lagrangian heuristics.
>
>
> The two problems on which we tested our method have indeed the same structure for their subproblems. We have not tested our approach on another problem but but this is work in progress. Thanks for pointing public available datasets and MAP problems on which we could try our approach.
>
> We have not compared our approach with non ML methods different from the bundle method. min-marginal averaging seems to be for solving ILPs but we have mixed integer linear problems.
>
>
> We do not have insights about the role that play coefficients in the GNN, but we reran the experiments without on the first fold of MCDN-SMALL-40:
>
> | Model   | GAP  | GAP-CR |
> | ------- | ---- | ------ |
> | w/coeff | 1.90 | 85.12  |
> | w/out coeff| 2.18 |83.32|
>
> As you can see omitting coefficient gives slightly better performance on both metrics.
>
>
> Code and instances will be available upon the acceptation of the paper.

---

### Official Review · Reviewer_Gopm · 2023-10-26

**Soundness:** 2 fair
**Presentation:** 3 good
**Contribution:** 2 fair
**Rating:** 5
**Confidence:** 3

**Summary:**

The authors propose a learning framework for computing good Lagrangian dual multipliers for solving mixed integer linear programs (MILPs). Numerical experiments on conducted on two MILP problems. The proposed method seems to provide Lagrangian multipliers that close much gap between the continuous relaxation bound and the optimal Lagrangian dual bound.

**Strengths:**

The proposed framework uses an architecture that can deal with variable input sizes. The proposed approach is tested on relevant MILP problems.

**Weaknesses:**

1. The technical contribution of the paper is very limited. Most techniques are from existing literature.
2. The numerical results are not strong enough. The proposed method does seem to be beneficial for obtaining an initial guess of the optimal dual. But it seems like the Lagrangian dual problem itself is not computationally hard (based on the results in Section 4) even on MCND-BIG-COMVAR. The optimal dual multipliers can be found easily by BM within a few minutes.
3. The writing can be improved. For example, CR is not defined (I assume it means continuous relaxation). I can find typos once in a while.

**Questions:**

It seems like the CR solution is important for learning a good dual solution. How does the learned dual solution compare with the CR dual solution in terms of GAP and GAP-CR?

---

> ### Author Response · Authors · 2023-11-15
>
> Thank your for you feedback.
>
> ### Contribution
> We are sorry to see that the main contribution was unclear.
> This work can be summarized as a modelling work rather than a technical work: the application of deep latent probabilistic modelling to MILP bound predictions. To our knowledge, this type of modelling is novel for this type of problem.
> Of course, there are already several works that base their models on encoder-decoder architectures, but they have a different goal, usually solving the primal problem with reinforcement learning.
>
> We believe that the topic and amount of content is in agreement with the ICLR call for paper.
>
> ### Numerical Results
>
> We agree that the optimal LMs can be obtained within 12 minutes for our more complex instances by BM. Still, this is quite long and could be replaced by a neural prediction, provided the accuracy is acceptable.
>
> For instance a simple k-NN cannot retrieve the correct LMs (on MCND-SMALL-40):
>
> | k   | GAP            | GAP-CR             |
> | --- | -------------- | ------------------ |
> | 3   | 58.12 (± 9.23) | -605.74 (± 527.38) |
> | 5   | 54.08 (± 8.51) | -556.23 (± 489.36) |
> | 10  | 52.32 (± 8.23) | -535.15 (± 473.33) |
>
> and using a MLP instead of our GNN-based probabilitic encoder and decoder is definitely not as good as our proposed method (and remember that it can only be applied to problems of the same size)
>
> | Obj       | GAP  | GAP-CR |
> | --------- | ---- | ------ |
> | LR (full) | 1.90 | 85.12  |
> | LR (MLP)  | 7.63 | 37.17  |
>
>
> ### Importance of GAP-CR and CR solution
> Yes the CR solution is important for learning a good dual solution.
> As shown in ablation studies, a system without this information (-cr) cannot predict accurate LMs.
>
> GAP-CR is exactly the metrics that  indicates how the learned dual solution compares with both the CR solution and the optimal solution.
> - a negative score indicates the predicted solution is worse than the CR
> - a zero score indicates that the solution is equivalent to CR
> - a 100 score is reached when the solution is equivalent to the one returned by BM

---

> > ### Comment · Reviewer_Gopm · 2023-11-15
> >
> > I would like to thank the authors for their response to my comments. I am not sure if the authors are answering the question I raised regarding the CR dual solution. Let me rephrase it below:
> > - There is a dual solution associated with CR (the optimal dual associated with Ax=b in the continuous relaxation). One can use that dual as Lagrangian multipliers to compute a bound, and I believe you have used it to initialize the solution for the bundle method. What are the GAP-CR and CR for that dual solution? I think it can be used as a benchmark because it is simple to compute.

---

> > > ### Author Response · Authors · 2023-11-17
> > >
> > > Thank you for your interest. Clearly, we had not understood your question.
> > > Using the dual variables $z^*$ associated with the dualized constraints as Lagrangian multipliers does not really improve the bound of the continuous relaxation CR. Here is a table indicating the CR gap closure given by $LR(z^*)$ on the different datasets (GAP-CR is equal to $(1 - \frac{LR(\pi^*) - LR(z^*)}{LR(\pi^*) - CR}) * 100$).
> > >
> > > |    Dataset       |GAP-CR |
> > > |:----------------:|------:|
> > > | MCND-SMALL_COM40 | 0.16% |
> > > | MCND-SMALL_COMVar| 0.06% |
> > > | MCND-BIG-COM40   | 0.61% |
> > > | MCND-BIG_COMVAR  | 0.28% |
> > > | CFL              | 0.67% |
> > >
> > > We will add this value in the paper for comparison with our prediction.

---

> > > > ### Comment · Reviewer_Gopm · 2023-11-20
> > > >
> > > > Thank you for the additional results. I have improved my score.

---

### Official Review · Reviewer_Qm6g · 2023-11-01

**Soundness:** 2 fair
**Presentation:** 3 good
**Contribution:** 2 fair
**Rating:** 3
**Confidence:** 4

**Summary:**

The paper considers mixed integer linear programs (MILPs). MILPs are NP-hard to solve optimally. A good approximation scheme is to use the Lagrangian to obtain good lower bounds. Hence, good Langrangian multipliers are needed for a specific MILP problem. The paper describes a deep learning approach based on a graph convolutional net to predict good Langrangian multipliers.

The paper also provides two sets of experiments that show the efficacy of the presented approach. In some cases (the Multi-Commodity Fixed-Charge Network Design Problem) it can close the gap between the continuous relaxation of the MILP and the best Lagrangian relaxation up to 85%. In others (the Capacitated Facility Location Problem) up to 50%.

**Strengths:**

The paper considers an important task of (approximately) solving MILPs by using the Lagrangian dual to obtain good lower bounds. The presented approach is sound and very interesting and seems to improve upon previous results in this area.

**Weaknesses:**

The paper considers a very important problem of finding good dual variables. While the presented approach seems plausible and useful, the paper is lacking a proper comparison to existing work. A good baseline that compares this approach over existing approaches is missing (in the experiments).  Also, it is unclear how the presented approach can really be beneficial. It is shown in the experiments that the network can predict good Lagrangian multipliers, such that a subsequent bundle method can be warm started and its iteration count is cut by one third. However, it would have been nice and essential to compare the running times also to state-of-the-art IP solvers like gurobi and also provide the instances and the code as a supplement such that they can be assessed by the reviewers.

Furthermore, it is not clear how well the approach really learns to predict the multipliers. If you provide enough training samples, like in your case, how well would a simple k-NN work?

Since MILPs are very important, and the presented approach is very general, it would have been nice to see it also applied to more general and more common MILPs. MCDN and CFL are somewhat special problems.

**Questions:**

1. How long does gurobi need to solve the MILP instances?
2. How does the approach compare to a simple k-NN baseline?
3. How does the approach compare to other approaches that learn Lagrangian dual variables? The paper states a number of such approaches for a number of specific MILPs.

---

> ### Author Response · Authors · 2023-11-15
>
> Thank you for your valuable feedback.
> We appreciate that you find our approach sound and very interesting.
> We agree that this is an important task and that we achieve good lower bounds.
>
>
> ### k-NN and Comparison with supervised methods / models
> We ran experiments to compare our approach with supervised methods (small MCDN instances, here evaluated on the first fold only)
> Optimal lagrangians are given by SMS++:
>
> | k   | GAP            | GAP-CR             |
> | --- | -------------- | ------------------ |
> | 3   | 58.12 (± 9.23) | -605.74 (± 527.38) |
> | 5   | 54.08 (± 8.51) | -556.23 (± 489.36) |
> | 10  | 52.32 (± 8.23) | -535.15 (± 473.33) |
>
> Clearly, simple k-NN is unable to retrieve interesting examples from the train data.
>
> Moreover we compare our model with other learning objective.
> We keep the encoder-decoder architecture of our model and change the objective for a supervised loss, either the Hinge Loss on the binary variable of each subproblem or more directly the MSE loss wrt to optimal LMs.
> In both cases, the "optimal solutions" are given by SMS++ and the evaluation is performed over the small MCDN instances, here evaluated on the first fold only:
>
> | Obj   | GAP  | GAP-CR |
> | ----- | ---- | ------ |
> | LR    | 1.90 | 85.12  |
> | Hinge | 3.99 | 65.57  |
> | MSE   | 9.10 | -3.16  |
>
> Unsupervised learning (LR) is clearly better than supervised losses.
> It should be noted that supervised learning is difficult, since there is an infinity of LM solutions that give the optimal bound.
> Learning to imitate one solver, here SMS++, by predicting  the same LMs is not an efficient learning method.
>
> If we replace the GNN encoder/decoder with a MLP, as done in methods where all problems have the same size, and train with the same LR objective we obtain (again, we train/evaluate on small instances with 40 commodities), we get:
>
> | Obj       | GAP  | GAP-CR |
> | --------- | ---- | ------ |
> | LR (full) | 1.90 | 85.12  |
> | LR (MLP)  | 7.63 | 37.17  |
>
> GNNs are essential to force the representation of the dualized constraints to agree with each other.
>
>
>
> ### MILP generic solvers and Choice of problems
>
> It is not clear how to compare Lagrangian Relaxation with an IP solver (Gurobi or CPLEX) since they do not compute the same thing. Lagrangian Relaxation gives a dual bound. Comparing the time to obtain this bound with the time necessary to solve to optimality does not seem fair. Using cplex/gurobi to compute the linear relaxation is quite fast and this is why we use its solution as features to predict a better bound.
>
>
> Our approach uses as features the primal and dual solutions of the continuous relaxation (CR) of the ILP on which LR is applied. Hence, our method is useful when:
> - LR provides a tighter bound than CR,
> - ILP contains a polynomial number of variables and constraints.
>
> We chose MCDN and CFL since they are well-known OR problems usually tackled by Lagrangian relaxation and satisfying the two mentionned requirements. On the contrary, the classic Lagrangian relaxation for TSP (one of the reference OR problems) based on one-tree decomposition does not satisfy any of our two requirements.
>
>
> ### Code and Data availability
>
> Code and instances will be available upon the acceptation of the paper.

---

### Official Review · Reviewer_F7ni · 2023-11-13

**Soundness:** 3 good
**Presentation:** 2 fair
**Contribution:** 1 poor
**Rating:** 3
**Confidence:** 3

**Summary:**

The authors presented an experiment report on solving a mixed integer linear program (MILP) by predicting the Lagrangian relaxation. They model the MILP problem by treating variable topology as a GNN (see [1] for an overview) and model the variable representation by an encoder-decoder architecture (this should be related to [2] despite not in an RL setup.) For prediction, they focus on the loss function by the Lagrangian relaxation with external convex relaxation input and predict the difference from the convex relaxation. Specifically, the draft take advantage of splitting the MILP problem by relaxing the harder constraints into the Lagrangian relaxation and using the exact solution of the easier problem from an outer solver as the training samples. Finally, the authors report their experiments on multi-commodity fixed-charge network design and capacitated facility location problems, and they report the ablation study on their solver variants.

[1] Combinatorial Optimization and Reasoning with Graph Neural Networks. Cappart et al. 2022
[2] Attention, Learn to Solve Routing Problems!. Kool et al. 2019

**Strengths:**

The draft looks more like an industrial, experimental report than a paper. The authors proved that the proposed method generalized well from the training dataset to the testing dataset. It has pretty good prediction errors in smaller datasets with one pass through the data and without RL in training. Further, the authors show that the learned solutions can warm-start the bundle methods. The solver may be valuable to the industry if the errors are acceptable.

**Weaknesses:**

However, the fatal benefit of the draft is that it doesn't include experimental comparisons to the other methods. Using DNN to improve combinatorial optimization has quite some literature, but the authors don't even cite [2], which has a close connection with the work on the encoder-decoder refinement in training. Further, in the experimental section, the authors only conduct experiments on self-generated datasets, which makes it even harder for outsiders to know what's happening. Thus, I can only recommend a rejection.

**Questions:**

1. P2: Please define the CR bound in your context.
2. P8 on the bundle method warm start. Does the time include the CR / DNN forward time?

---

> ### Author Response · Authors · 2023-11-15
>
> Thank you for your feedback. We appreciate that you find that we obtain good prediction errors with our prediction model and that our method generalizes well. We also appreciate that  we convince you that our method is valuable for warmstarting bundle methods.
>
> Thank you for pointing out reference [2].
> We agree that it is a really great work that we could have cited in our paper. We did not mention it since our work is related to predicting dual bound, and not primal bound as in [2]. We cite instead (Nair et al.) when mentionning primal bound prediction as their model is on predicting primal bounds of any MILP, and not vehicle routing like in [2]. Moreover, we do not use Reinforcement learning as in [2].
> Still, we will take you feedback into account and cite it as a previous work on encoder-decoder architectures for optimization-based problems.
>
> We could not find datasets available for these problems, that is why we had to create new ones.
> Upon acceptance, we will release our datasets to encourage comparisons and further work.
>
>
> Q1/2: The results on page 8 indeed include the time for solving the continuous relaxation (CR) and making the prediction with our ML model. We will make it clearer in the revised version.

---

### Meta-Review · Area_Chair_rAju · 2023-12-06

**Metareview:**

This submission introduces a deep learning method for predicting Lagrangian multipliers in Mixed Integer Linear Programs (MILP). This approach, leveraging a graph convolutional network, marks a departure from traditional iterative algorithms, aiming to reduce computational complexity and improve efficiency in solving MILPs with complex constraints. The practical application of this method is demonstrated in multi-commodity fixed-charge network design and capacitated facility location problems.

Despite its innovation, a fatal shortcoming of this paper is its lack of comparisons with existing methods. This is particularly problematic in view of the fact that the basic idea has been approached many times before, and so the proposed method must not only perform well in a vacuum, but also outperform existing baselines. Additionally, the experiments rely on self-generated datasets, raising questions about the generalizability and robustness of the method. Reviewers pointed out the need for a broader range of problem tests and benchmarks, such as comparisons with traditional solvers and simpler approaches like k-NN.

**Justification For Why Not Higher Score:**

The weaknesses outweighed the strengths of the paper.

**Justification For Why Not Lower Score:**

N/A

---

### Decision · Program_Chairs · 2024-01-16

Reject